# Quantitative radiomics approach to assess acute radiation dermatitis in breast cancer patients

**So-Yeon Park[1,2], Jong Min Park[2,3,4,5], Jung-in Kim[2,3,4], Chang Heon Choi[2,3,4], Minsoo Chun[2,6], Ji Hyun Chang[2,3,4], Jin Ho Kim[2,3,4]***

1 Department of Radiation Oncology, Veterans Health Service Medical Center, Seoul, Republic of Korea, 2 Institute of Radiation Medicine, Seoul National University Medical Research Center, Seoul, Republic of Korea, 3 Department of Radiation Oncology, Seoul National University Hospital, Seoul, Republic of Korea, 4 Biomedical Research Institute, Seoul National University Hospital, Seoul, Republic of Korea, 5 Center for Convergence Research on Robotics, Advanced Institutes of Convergence Technology, Suwon, Republic of Korea, 6 Department of Radiation Oncology, Chung-Ang University Hospital, Seoul, Republic of Korea

* jinho.kim.md@gmail.com

## Abstract

**Data Availability Statement:** All relevant data are within the paper and its Supporting Information files.

### Purpose

We applied a radiomics approach to skin surface images to objectively assess acute radiation dermatitis in patients undergoing radiotherapy for breast cancer.

### Methods

A prospective cohort study of 20 patients was conducted. Skin surface images in normal, polarized, and ultraviolet (UV) modes were acquired using a skin analysis device before starting radiotherapy ('Before RT'), approximately 7 days after the first treatment ('RT D7'), on 'RT D14', and approximately 10 days after the radiotherapy ended ('After RT D10'). Eighteen types of radiomic feature ratios were calculated based on the values acquired 'Before RT'. We measured skin doses in ipsilateral breasts using optically stimulated luminescent dosimeters on the first day of radiotherapy. Clinical evaluation of acute radiation dermatitis was performed using the Radiation Therapy Oncology Group scoring criteria on 'RT D14' and 'After RT D10'. Several statistical analysis methods were used in this study to test the performance of radiomic features as indicators of radiodermatitis evaluation.

### Results

As the skin was damaged by radiation, the *energy* for normal mode and *sum variance* for polarized and UV modes decreased significantly for ipsilateral breasts, whereas contralateral breasts exhibited a smaller decrease with statistical significance. The radiomic feature ratios at 'RT D7' had strong correlations to skin doses and those at 'RT D14' and 'after RT D10' with statistical significance.

**Funding:** This study was supported by Basic Science Research Program through the National Research Foundation of Korea (NRF) funded by the Ministry of Education (NRF-2017R1D1A1B03036093) and by the Korea government (MSIT) (No. RS-2023-00253604). The funders had no role in study design, data collection and analysis, decision to publish, or preparation of the manuscript.

**Competing interests:** The authors have declared that no competing interests exist.

## Conclusions

The *energy* for normal mode and *sum variance* for polarized and UV modes demonstrated the potential to evaluate and predict acute radiation, which assists in its appropriate management.

## Introduction

Acute radiation dermatitis is a common side effect in patients undergoing radiotherapy for breast cancer. The physical manifestations of radiodermatitis can range from faint erythema and desquamation to skin necrosis and ulceration, which impacts a patient's physical condition and quality of life [1–5]. Risk factors for radiodermatitis include both treatment- and patient-related factors; however, they primarily depend on skin doses, which are classified as the treatment-related factor [6–8]. Depending on its severity, acute radiation dermatitis may cause an interruption in or cessation of radiotherapy, which may lead to poor clinical outcomes [9–14].

Acute radiation reactions of the skin occur due to damage to the basal cell layer of the skin, which leads to an imbalance between the normal production of cells in this layer and the destruction of cells at the skin surface [15]. It appears 10–14 days from the first fraction of radiotherapy, corresponding with the time it takes for the damaged basal cells to migrate to the skin surface [16, 17]. Because the skin is further damaged by radiation during radiotherapy, its reactions increase in severity and peak approximately 7–10 days after treatment has completed [2]. Subsequently, the acute radiation dermatitis will start to subside and gradually heal [18, 19].

Traditionally, the severity of acute radiation dermatitis is still subjectively evaluated and graded by the clinician using scoring criteria, such as those provided by the Radiation Therapy Oncology Group (RTOG) [17, 20, 21], the Common Terminology Criteria for Adverse Events [22, 23], and the World Health Organization criteria [24–26]. The inherent subjectivity of these criteria can cause high inter-observer variability, and they do not provide quantitative information for some clinical biomarkers. Another limitation is the diversity in the types of radiation damage that present the same physical manifestation on the skin (e.g., faint erythema); all of these types may be simply categorised under the same severity grade. Consequently, several 'objective' methods to quantify changes in the acute radiation reaction of the skin have been introduced by utilising a spectrophotometer, corneometer, and laser Doppler flowmetry [19, 27–31]. Schmeel *et al.* provided a reliable and precise measure of skin severity, based on a three-dimensional colour system, which consists of L* (lightness), a* (colour ranging from red to green), and b* (colour ranging from blue to yellow) [31]. Because radiation makes the skin darker and redder, the L* values decreased and the a* values increased, whereas changes in b* values were negligible. Kawamura *et al.* generated an objective scoring system to predict acute radiation dermatitis in patients treated with radiotherapy for head and neck cancer using dose-volumetric parameters of the skin [7].

Radiomics is a new concept that can be used to provide quantitative measures of tumours or normal tissues by analysing medical images from various modalities; it has attracted considerable interest in the field of radiology. Radiomic features from several non-invasive medical images can be utilised by integrating clinical biomarkers, genetic and biochemical indices, and treatment outcomes into a sophisticated model to improve diagnostic assessment and prognostic determinants of patients [32, 33]. Recently, the clinical application of radiomics has

expanded from internal image modalities, which image the internal body parts, such as computed tomography (CT), magnetic resonance imaging (MRI), and ultrasonography (US), to external image modalities, which scan the skin surface, such as a thermal imaging device and digital cameras. Several previous studies have shown that the radiomic features of skin surface images acquired by these modalities have the potential to monitor or predict the skin severity [34–36]. Although the feasibility of thermal and digital skin imaging for the whole chest area to evaluate the occurrence of acute radiation dermatitis has been demonstrated in previous studies, there were limitations in analysing the large area image of the whole breast, resulting in a low-resolution bias. Moreover, previous studies have investigated the performance of these objective methods by correlating the 'subjective' scoring criteria.

In our study, we evaluated the severity of acute radiation dermatitis for patients with breast cancer by applying a radiomics approach to topical images measured by a skin analysis device. The skin analysis device acquires images of the skin surface with the highest resolution, which can provide qualitatively and quantitatively sufficient information for radiomics analysis. We analysed the correlation between the relative changes in radiomic features calculated from the longitudinal images (defined as 'delta-radiomics') and measured skin doses. We propose an alternative objective method to assess the skin severity at early treatment time intervals with longitudinal assessment.

## Materials and methods

### Patients

This is a prospective cohort study conducted in a single centre between October 2018 and September 2020. Female patients with newly diagnosed ductal carcinoma in situ were enrolled. All patients underwent breast-conserving surgery followed by whole-breast radiotherapy. The exclusion criteria were as follows: age > 70 years; prior history of radiotherapy for breast; recurrent breast cancer; bilateral breast cancer; patients with distant metastasis from breast cancer; previous breast reconstruction or implants; and patients receiving previous or simultaneous chemotherapy. This study was approved by the Institutional Review Board of Seoul National University hospital (IRB No. D-1810-053-977) and was conducted according to the Declaration of Helsinki. Informed written consent was obtained from all patients prior to their participation in this study.

### Radiotherapy

All patients underwent CT scans using the Brilliance CT Big Bore™ (Philips, Amsterdam, Netherlands) with a slice thickness of 3 mm. The patients were immobilised on a breast board (CIVCO Radiotherapy, Coralville, IA, USA) in supine position with both arms raised above their heads. The target volume was defined as the whole ipsilateral breast. The patients received 40.5 Gy in 15 fractions using 6 MV, or a combination of 6 MV and 10 MV, photons. Depending on the side of the breast cancer treatment, tangential photon beams as intensity modulated radiotherapy (IMRT) or field-in-field (FIF) techniques were generated for left- or right-sided breast cancers, respectively. At least 95% of the target volume should be covered by 100% of the prescription dose. The maximum dose to the target volume did not typically exceed 115% of the prescription dose. For all patients, boost irradiation was not administered.

### Clinical assessment of acute radiation dermatitis using scoring criteria

Clinical evaluation of acute radiation dermatitis using RTOG scoring criteria [20] was performed approximately 14 days after the first day of radiotherapy ('RT D14'). The subsequent

assessment was performed approximately 10 days after the end of the radiotherapy ('After RT D10'). All patients were assessed independently by an experienced radiation oncologist who was blind to the results of both the radiomics analysis and skin doses. When the evaluation session overlapped with holidays or weekend, we evaluated weekday sessions within a 3-day window.

## Skin dose measurement

The nanoDot optically stimulated luminescent dosimeter (OSLD) system (Landauer Inc., Glenwood, IL, USA) was used to measure skin doses to ipsilateral breasts of patients on the first day of radiotherapy. To reduce the dose measurement uncertainty, we used OSLDs prepared with pre-irradiation with dose values greater than 5 kGy [37]. For the calibration of pre-irradiated OSLDs, the absolute dose of linear accelerator was calibrated according to the American Association of Physicists in Medicine Task Group 51 protocol [38]. Subsequently, dose values ranging from 1 to 10 Gy were delivered to all sets of OSLD. The measurement setup consisted of a depth of maximum dose, an SSD of 100 cm and a field size of $10 \times 10$ cm$^2$. Four measurement points were placed on the upper, lower, inner, and outer sides of the ipsilateral breast. Each point was located at a distance of 3 cm from the nipple of the ipsilateral breast. The reason for selecting the nipple as the reference point for measurements was to position the OSLDs several times in a reproducible manner among the skin. One set of three OSLDs was utilised for each measurement point and the dose values of those OSLDs were then averaged. Surgical scars were avoided when selecting the points.

## Acquisition of skin surface image with skin analysis device and radiomics analysis

Skin surface imaging was performed at room temperature ($25 \pm 1˚$C) and under ambient room fluorescent lights using a portable skin analysis device (API-100; Aram Huvis, Gyeonggi-do, Korea). This device has a measurement area of $1 \times 1$ cm$^2$ with a resolution of $1624 \times 1212$ pixels and it provides three different modes of skin surface imaging: normal, polarized, and ultraviolet (UV) modes. For the acquisition of skin surface images, all patients were positioned, while lying on their back in a couch, with their arms raised above their heads. All measurements were performed by the same investigator. No topical products, including cosmetics, emollient, and cleansers, were permitted for at least 8 h before the skin measurement [29]. Skin surface images were acquired for the ipsilateral breasts in three modes at the corresponding points measured by OSLDs at four different times. The measurement times were before starting radiotherapy ('Before RT'), 'RT D7,' 'RT D14,' and 'After RT D10'. For comparison, the corresponding points of the contralateral breast were measured in the same manner as the controls. The total measurement time was less 10 min. A timeline of our experimental design is depicted in Fig 1.

The acquired image was in a red-green-blue (RGB) colour format with 24 bits per pixel, and it was separated into three channels for analysis. For the red channel, each image was cropped into a circular region-of-interest (ROI) in the centre of the image with a radius of 300 pixels to avoid the analysis of unnecessary portions of the image. To extract radiomic features from the image, the cropped images were analysed using second-order statistics based on a grey level co-occurrence matrix (GLCM) [39]. To generate the GLCM, the number of quantised grey levels ($N_g$), the value of the displacement vector ($d$), and the values of the offset angles ($\theta$) were set to $N_g = 64$, $d = 1$, and $\theta = 0˚$, $45˚$, $90˚$, and $135˚$, respectively. A total of 18 radiomic features were calculated based on the acquired GLCMs using open-source code in MATLAB software (Mathworks, Natick, MA, USA) [40]. For each image, a total of four radiomic features were calculated according to the four offset angles, and those features were then

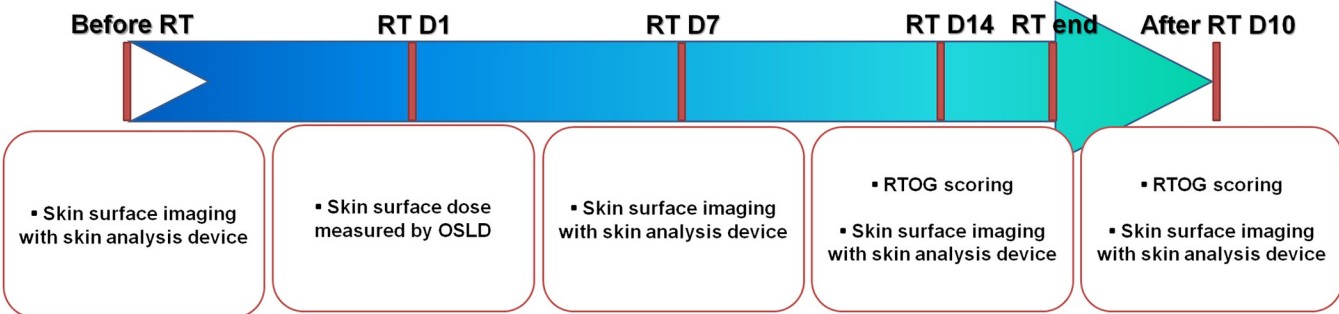

**Fig 1. Timeline of the experimental design of this study.** 'Before RT,' 'RT D*n*,' 'RT end,' and 'After RT D*n*' refer to 'before starting radiotherapy,' 'approximately *n* days after the first day of radiotherapy,' 'at the end of radiotherapy,' and approximately *n* days after the end of radiotherapy,' respectively, except that 'RT D1' refers to the first day of radiotherapy.

averaged for each image. For the green and blue channels, we calculated the radiomic features in the same manner described above. The three radiomic features from each channel were averaged for each original image in the three channels. Thus, for each patient, a total of 1,728 radiomic features (18 radiomic features × four measurement points × both breasts × four time points × three imaging modes) were calculated in this study.

For each breast side, delta-radiomics defined as the ratio of the values of the radiomic features for all time points ('Before RT,' 'RT D7,' 'RT D14,' and 'After RT D10') to those for the first time point ('Before RT') were calculated to consider the diversity of colour, tone, and texture of the skin among individuals.

## Statistical analysis

A two-way repeated measures analysis of variance (ANOVA) was used to test the significant changes in the delta-radiomics within each group (ipsilateral or contralateral breast) across time, and the significant differences in those for the interaction of time and group. If the Mauchly's test of sphericity assumption was violated, the Greenhous–Geisser correction was applied. $P$-values less than 0.05 were considered as statistically significant. Depending on the results of the Shapiro–Wilk test for the normality of the two corresponding datasets, a paired-t test or Wilcoxon signed rank test was used for pairwise comparisons of the delta-radiomics between ipsilateral and contralateral breasts for each time point. Bonferroni's correction, which considers the adjusted significance level of 0.0167, was applied for *post hoc* comparisons. Correlations between skin doses and the delta-radiomics for ipsilateral breasts for all time points were determined using Pearson's or Spearman's correlation test for parametric or non-parametric data, respectively. For ipsilateral breasts, Pearson's or Spearman's correlation coefficients were calculated to assess the correlations of the delta-radiomics for 'RT D7' to those for 'RT D14' and 'After RT D10,' depending on the normality of the data. For these statistical tests, correction for multiple comparisons was performed with the Benjamin-Hochberg method to limit the false discovery rate at the 5% level. The adjusted $p$-values less than 0.05 were considered as statistically significant. All statistical analyses were performed using PASW statistic 18.0 software (SPSS, Chicago, IL, USA).

## Results

### Patients and RTOG scoring

A total of 20 patients were enrolled in this prospective study. Table 1 shows the clinical characteristics of the patients, including age, tumour location, plan techniques and photon energies,

**Table 1. Patient characteristics (n = 20).**

| Characteristics | Case number (%) |
|---|---|
| Median age (range, years) | 49.5 years (40–68) |
| Tumour location | |
| Left | 8 (40%) |
| Right | 12 (60%) |
| Plan technique | |
| FIF | 12 (60%) |
| IMRT | 8 (40%) |
| Energy | |
| 6 MV | 17 (85%) |
| 10 MV | 1 (5%) |
| 6 + 10 MV | 2 (10%) |
| Chemotherapy | |
| Yes | 0 (0%) |
| No | 20 (100%) |
| Hormone therapy | |
| Yes | 15 (75%) |
| No | 5 (25%) |

*Abbreviations*: FIF = field-in-field; IMRT = intensity modulated radiotherapy.

and different treatments for breast cancer (previous chemotherapy and hormone therapy). For all patients, the pathologic stage of breast cancer was pTisN0M0.

Based on the RTOG scoring criteria, all patients were classified as grade 1 at the 'RT D14' and 'After RT D10' times.

## Skin dose measurement

The values measured using the OSLD dosimeters are summarised in S1 Table. The mean values of the skin doses for the upper, lower, inner, and outer sides of the ipsilateral breast were 253.1 ± 14.7, 247.8 ± 18.1, 212.4 ± 19.6, and 226.8 ± 23.5 cGy, respectively. Most of the mean dose values for the inner ipsilateral breast (14 cases; 70%) were the lowest. The mean dose values for the upper side of the ipsilateral breast were comparable to those for the lower side.

## Longitudinal assessment of changes in skin colour and texture by radiotherapy

We examined the changes in skin colour and texture according to the delta-radiomics of the skin surface image measured using a skin analysis device. Fig 2 shows the skin colour and texture changes and the corresponding GLCMs over time in the three imaging modes. There were noticeable changes in the skin surface images, owing to radiation damage. The calculated delta-radiomics for ipsilateral and contralateral breasts in three imaging modes (normal, polarized, and UV lights) are presented in Tables 2–4, respectively. *P*-values obtained using the two-way repeated measures ANOVA and paired-t tests (or Wilcoxon signed rank tests) are shown in S2 and S3 Tables, respectively.

In the normal imaging mode, all delta-radiomics exhibited statistically significant changes within and between groups (ipsilateral and contralateral breasts) over time. Among all features, the *energy*, *entropy*, *homogeneity*, and *sum entropy* exhibited differences in the delta-

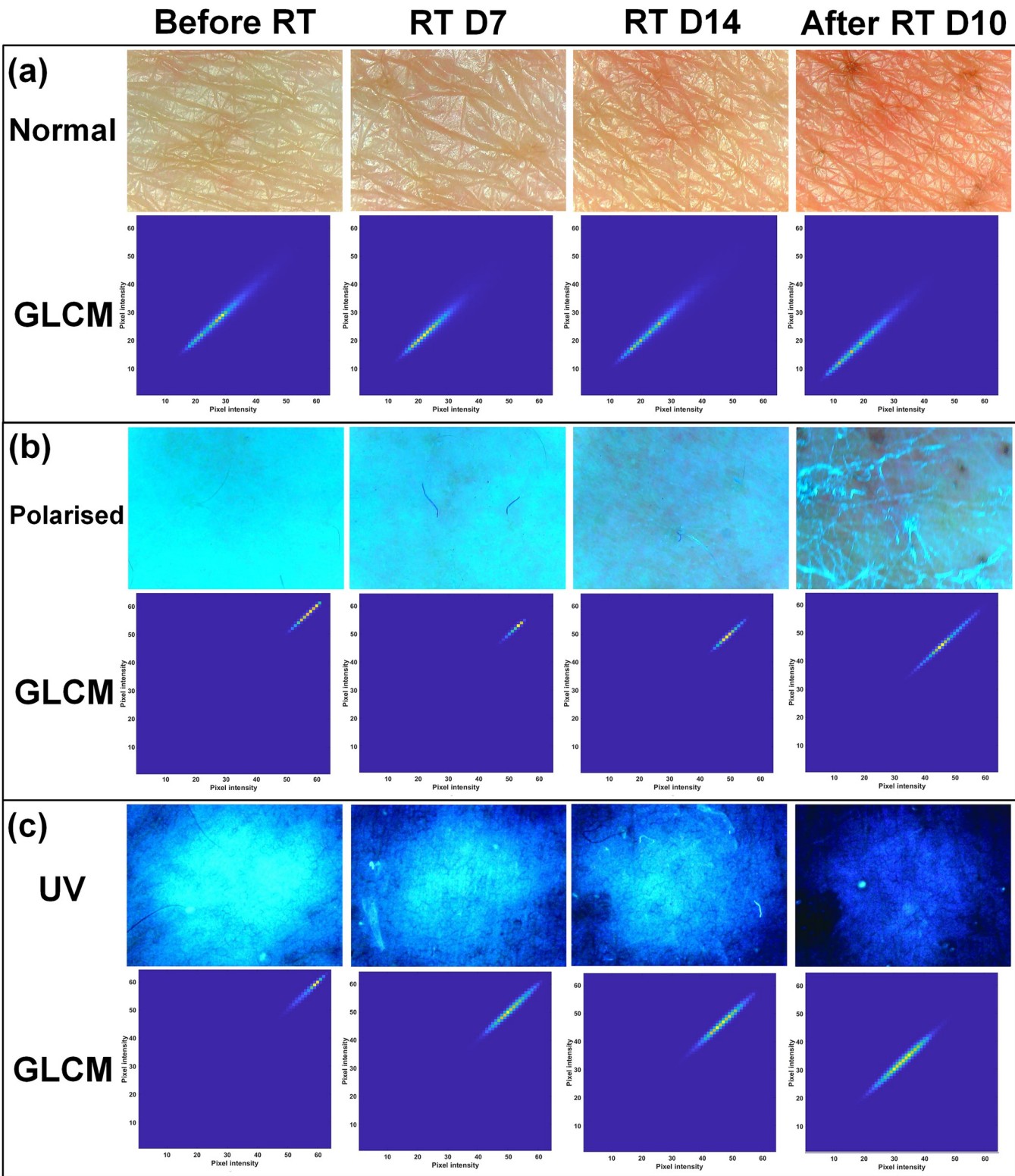

**Fig 2.** Skin surface images and the corresponding grey level co-occurrence matrices (GLCMs) over time in the (a) normal, (b) polarized, and (c) UV modes. The skin image measurements were performed before starting the radiotherapy ('Before RT'), approximately 7 days after the first day of the radiotherapy ('RT D7'), on 'RT D14,' and approximately 10 days after the end of the radiotherapy ('After RT D10').

**Table 2. The mean and standard deviation of delta-radiomics for ipsilateral and contralateral breasts in the normal imaging mode.**

| | Ipsilateral breast | | | | Contralateral breast | | | |
|---|---|---|---|---|---|---|---|---|
| | Before RT | RT D7 | RT D14 | After RT D10 | Before RT | RT D7 | RT D14 | After RT D10 |
| Autocorrelation* | 1.000 ± 0.000 | 0.966 ± 0.054 | 0.961 ± 0.036 | 1.008 ± 0.060 | 1.000 ± 0.000 | 0.990 ± 0.045 | 0.990 ± 0.046 | 1.002 ± 0.048 |
| Contrast* | 1.000 ± 0.000 | 1.562 ± 0.624 | 1.979 ± 0.965 | 1.207 ± 0.583 | 1.000 ± 0.000 | 1.042 ± 0.358 | 1.078 ± 0.522 | 1.065 ± 0.498 |
| Correlation* | 1.000 ± 0.000 | 0.992 ± 0.012 | 0.987 ± 0.015 | 1.002 ± 0.013 | 1.000 ± 0.000 | 1.002 ± 0.011 | 1.001 ± 0.015 | 1.002 ± 0.016 |
| Cluster prominence* | 1.000 ± 0.000 | 1.789 ± 0.670 | 2.254 ± 1.173 | 1.334 ± 0.667 | 1.000 ± 0.000 | 1.230 ± 0.497 | 1.169 ± 0.497 | 1.078 ± 0.560 |
| Cluster shade* | 1.000 ± 0.000 | 2.108 ± 1.083 | 2.508 ± 1.443 | 1.111 ± 1.235 | 1.000 ± 0.000 | 1.658 ± 1.622 | 1.540 ± 1.375 | 1.487 ± 1.902 |
| Dissimilarity* | 1.000 ± 0.000 | 1.221 ± 0.260 | 1.399 ± 0.356 | 1.121 ± 0.301 | 1.000 ± 0.000 | 0.988 ± 0.185 | 0.984 ± 0.241 | 0.995 ± 0.261 |
| Energy*, † | 1.000 ± 0.000 | 0.848 ± 0.109 | 0.775 ± 0.140 | 0.672 ± 0.155 | 1.000 ± 0.000 | 1.072 ± 0.164 | 1.107 ± 0.191 | 1.072 ± 0.235 |
| Entropy*, † | 1.000 ± 0.000 | 1.045 ± 0.052 | 1.084 ± 0.066 | 1.039 ± 0.066 | 1.000 ± 0.000 | 0.994 ± 0.048 | 0.988 ± 0.057 | 0.989 ± 0.065 |
| Homogeneity*, † | 1.000 ± 0.000 | 0.966 ± 0.049 | 0.933 ± 0.060 | 0.973 ± 0.065 | 1.000 ± 0.000 | 1.011 ± 0.042 | 1.017 ± 0.055 | 1.012 ± 0.068 |
| Maximum probability* | 1.000 ± 0.000 | 0.971 ± 0.183 | 0.886 ± 0.166 | 0.844 ± 0.161 | 1.000 ± 0.000 | 1.073 ± 0.138 | 1.101 ± 0.146 | 1.045 ± 0.173 |
| Variance* | 1.000 ± 0.000 | 0.967 ± 0.054 | 0.961 ± 0.036 | 1.008 ± 0.060 | 1.000 ± 0.000 | 0.990 ± 0.044 | 0.990 ± 0.046 | 1.002 ± 0.048 |
| Sum average* | 1.000 ± 0.000 | 0.982 ± 0.027 | 0.979 ± 0.019 | 1.003 ± 0.030 | 1.000 ± 0.000 | 0.994 ± 0.023 | 0.995 ± 0.023 | 1.001 ± 0.024 |
| Sum variance* | 1.000 ± 0.000 | 0.964 ± 0.056 | 0.957 ± 0.037 | 1.006 ± 0.063 | 1.000 ± 0.000 | 0.990 ± 0.047 | 0.990 ± 0.049 | 1.003 ± 0.051 |
| Sum entropy*, † | 1.000 ± 0.000 | 1.020 ± 0.026 | 1.037 ± 0.029 | 1.028 ± 0.027 | 1.000 ± 0.000 | 0.999 ± 0.026 | 0.994 ± 0.025 | 0.994 ± 0.029 |
| Difference variance* | 1.000 ± 0.000 | 1.562 ± 0.624 | 1.979 ± 0.965 | 1.207 ± 0.583 | 1.000 ± 0.000 | 1.042 ± 0.358 | 1.078 ± 0.522 | 1.065 ± 0.498 |
| Difference entropy* | 1.000 ± 0.000 | 1.121 ± 0.143 | 1.210 ± 0.170 | 1.046 ± 0.171 | 1.000 ± 0.000 | 0.993 ± 0.120 | 0.988 ± 0.147 | 0.982 ± 0.154 |
| IMC 1* | 1.000 ± 0.000 | 0.945 ± 0.091 | 0.893 ± 0.100 | 0.984 ± 0.116 | 1.000 ± 0.000 | 1.024 ± 0.086 | 1.030 ± 0.113 | 1.028 ± 0.134 |
| IMC 2* | 1.000 ± 0.000 | 0.996 ± 0.009 | 0.992 ± 0.011 | 1.000 ± 0.010 | 1.000 ± 0.000 | 1.002 ± 0.007 | 1.002 ± 0.010 | 1.001 ± 0.012 |

*Abbreviations*: Before RT = before starting the radiotherapy; RT D*n* = approximately *n* days after the first day of the radiotherapy; After RT D*n* = approximately *n* days after the end of the radiotherapy; IMC = inverse measure of correlation.

*indicates statistical significances ($p < 0.05$) of both changes in the delta-radiomics within each group across time and differences in those for the interaction of the time and group with a two-way repeated measures analysis of variance (ANOVA).

†indicates statistical significances (p-value $< 0.05/3 = 0.0167$) of differences in the delta-radiomics between ipsilateral and contralateral breasts for all time points with a paired-t test or Wilcoxon signed rank test, depending on the normality of data.

radiomics between ipsilateral and contralateral breasts for all time points with statistical significance. Every delta-radiomics for ipsilateral breasts tended to increase or decrease until 'RT D14.' However, they exhibited the opposite trend 'After RT D10,' except for the *energy* and *maximum probability*, which decreased significantly. While all delta-radiomics for ipsilateral breasts showed noticeable tendencies over time, those for contralateral breasts exhibited unpredictable tendencies with smaller variations over time.

In the polarized imaging mode, every delta-radiomics exhibited statistically significant changes within and between groups across time, except for the *correlation*, *cluster prominence*, *cluster shade*, *maximum probability*, and *inverse measure of correlation 2 (IMC 2)*. Among them, most of the features also showed statistically significant differences in the delta-radiomics values between the groups for all time points, except for the *contrast*, *energy*, *sum entropy*, and *difference variance*. For the statistically significant delta-radiomics of ipsilateral breasts, the *dissimilarity*, *entropy*, *sum entropy*, and *difference entropy* increased, while the *energy*, *homogeneity*, and *sum variance* decreased during the time course. The corresponding delta-radiomics for contralateral breasts exhibited the same trends as those for ipsilateral breasts; however, its change width was smaller than that for ipsilateral breasts.

In the UV imaging mode, of the 12 radiomic features that showed significant differences for the interaction of time and group, the *autocorrelation*, *variance*, *sum average*, and *sum variance* showed statistically significant differences between groups for all time points. Every radiomic feature for the ipsilateral breasts exhibited increasing or decreasing tendencies over

**Table 3. The mean and standard deviation of delta-radiomics for ipsilateral and contralateral breasts in the polarized imaging mode.**

| | Ipsilateral breast | | | | Contralateral breast | | | |
|---|---|---|---|---|---|---|---|---|
| | Before RT | RT D7 | RT D14 | After RT D10 | Before RT | RT D7 | RT D14 | After RT D10 |
| Autocorrelation*, † | 1.000 ± 0.000 | 0.936 ± 0.062 | 0.879 ± 0.098 | 0.904 ± 0.115 | 1.000 ± 0.000 | 0.973 ± 0.043 | 0.961 ± 0.060 | 0.954 ± 0.072 |
| Contrast* | 1.000 ± 0.000 | 1.458 ± 0.602 | 1.931 ± 0.737 | 1.858 ± 1.358 | 1.000 ± 0.000 | 1.139 ± 0.330 | 1.165 ± 0.348 | 1.309 ± 0.469 |
| Correlation | 1.000 ± 0.000 | 0.997 ± 0.009 | 0.998 ± 0.009 | 1.003 ± 0.013 | 1.000 ± 0.000 | 1.000 ± 0.008 | 1.003 ± 0.008 | 1.003 ± 0.008 |
| Cluster prominence | 1.000 ± 0.000 | 2.779 ± 3.898 | 4.773 ± 7.285 | 18.476 ± 56.921 | 1.000 ± 0.000 | 1.962 ± 2.764 | 2.076 ± 2.317 | 3.451 ± 5.296 |
| Cluster shade | 1.000 ± 0.000 | 1.100 ± 3.978 | 4.062 ± 8.047 | 18.125 ± 56.285 | 1.000 ± 0.000 | 1.414 ± 3.313 | 1.168 ± 2.299 | 2.042 ± 5.708 |
| Dissimilarity*, † | 1.000 ± 0.000 | 1.274 ± 0.325 | 1.590 ± 0.333 | 1.638 ± 0.402 | 1.000 ± 0.000 | 1.079 ± 0.246 | 1.151 ± 0.249 | 1.251 ± 0.306 |
| Energy* | 1.000 ± 0.000 | 0.902 ± 0.205 | 0.713 ± 0.169 | 0.621 ± 0.165 | 1.000 ± 0.000 | 0.984 ± 0.252 | 0.883 ± 0.284 | 0.831 ± 0.254 |
| Entropy*, † | 1.000 ± 0.000 | 1.074 ± 0.133 | 1.175 ± 0.130 | 1.228 ± 0.137 | 1.000 ± 0.000 | 1.031 ± 0.134 | 1.074 ± 0.142 | 1.103 ± 0.140 |
| Homogeneity*, † | 1.000 ± 0.000 | 0.983 ± 0.017 | 0.962 ± 0.015 | 0.952 ± 0.030 | 1.000 ± 0.000 | 0.996 ± 0.014 | 0.989 ± 0.017 | 0.982 ± 0.024 |
| Maximum probability | 1.000 ± 0.000 | 0.931 ± 0.205 | 0.778 ± 0.170 | 0.703 ± 0.166 | 1.000 ± 0.000 | 1.011 ± 0.254 | 0.915 ± 0.266 | 0.869 ± 0.246 |
| Variance*, † | 1.000 ± 0.000 | 0.936 ± 0.062 | 0.879 ± 0.098 | 0.904 ± 0.115 | 1.000 ± 0.000 | 0.973 ± 0.042 | 0.961 ± 0.060 | 0.954 ± 0.072 |
| Sum average*, † | 1.000 ± 0.000 | 0.967 ± 0.033 | 0.935 ± 0.055 | 0.946 ± 0.067 | 1.000 ± 0.000 | 0.986 ± 0.022 | 0.980 ± 0.032 | 0.975 ± 0.039 |
| Sum variance*, † | 1.000 ± 0.000 | 0.920 ± 0.059 | 0.859 ± 0.093 | 0.795 ± 0.105 | 1.000 ± 0.000 | 0.971 ± 0.047 | 0.958 ± 0.065 | 0.949 ± 0.077 |
| Sum entropy* | 1.000 ± 0.000 | 1.061 ± 0.126 | 1.149 ± 0.122 | 1.203 ± 0.131 | 1.000 ± 0.000 | 1.028 ± 0.130 | 1.070 ± 0.137 | 1.094 ± 0.131 |
| Difference variance* | 1.000 ± 0.000 | 1.458 ± 0.602 | 1.931 ± 0.737 | 1.858 ± 1.358 | 1.000 ± 0.000 | 1.139 ± 0.330 | 1.165 ± 0.348 | 1.309 ± 0.469 |
| Difference entropy*, † | 1.000 ± 0.000 | 1.162 ± 0.179 | 1.341 ± 0.180 | 1.347 ± 0.196 | 1.000 ± 0.000 | 1.050 ± 0.148 | 1.092 ± 0.144 | 1.146 ± 0.169 |
| IMC 1*, † | 1.000 ± 0.000 | 0.962 ± 0.030 | 0.930 ± 0.025 | 0.937 ± 0.056 | 1.000 ± 0.000 | 0.994 ± 0.018 | 0.989 ± 0.021 | 0.978 ± 0.032 |
| IMC 2 | 1.000 ± 0.000 | 1.000 ± 0.008 | 1.003 ± 0.007 | 1.006 ± 0.010 | 1.000 ± 0.000 | 1.001 ± 0.009 | 1.003 ± 0.010 | 1.004 ± 0.009 |

*Abbreviations*: Before RT = before starting the radiotherapy; RT D*n* = approximately *n* days after the first day of the radiotherapy; After RT D*n* = approximately *n* days after the end of the radiotherapy; IMC = inverse measure of correlation.

*indicates statistical significances ($p < 0.05$) of both changes in the delta-radiomics within each group across time and differences in those for the interaction of the time and group with a two-way repeated measures analysis of variance (ANOVA).

†indicates statistical significances (p-value $< 0.05/3 = 0.0167$) of differences in the delta-radiomics between ipsilateral and contralateral breasts for all time points with a paired-t test or Wilcoxon signed rank test, depending on the normality of data.

time, except the *cluster shade*, *maximum probability*, and *IMC 1*. Similar to the polarized imaging mode, the corresponding radiomic features for contralateral breasts showed the same trends as those for the ipsilateral breast; however, its change width was smaller than that for ipsilateral breasts. Overall, changes between time points in the UV imaging mode were larger than those in the normal and polarized imaging modes.

No significant relationship between the delta-radiomics for ipsilateral breasts and RTOG scoring was observed.

## Correlations of delta-radiomics with skin doses

For this analysis, the delta-radiomics for the ipsilateral breasts were used. The values of the correlation coefficient (*r*), and the corresponding *p*-values for the correlations between the skin doses, and the delta-radiomics at 'RT D7,' 'RT D14,' and 'After RT D10' were calculated. No significant correlations between the delta-radiomics at 'RT D14' and 'After RT D10' and the skin doses were identified. The values of *r* and corresponding *p*-values of skin doses and delta-radiomics at 'RT D7' are shown in Table 5. Only the *r* values with *p*-values less than 0.05 are listed in Table 5.

In the normal imaging mode, radiomic features with *p*-values less than 0.05 were the *energy*, *entropy*, *homogeneity*, and *maximum probability*. The *energy* had a significantly higher value of *r* compared with other radiomic features to skin doses. In the polarized imaging mode, eight

**Table 4. The mean and standard deviation of delta-radiomics for ipsilateral and contralateral breasts in the UV imaging mode.**

| | Ipsilateral breast | | | | Contralateral breast | | | |
|---|---|---|---|---|---|---|---|---|
| | Before RT | RT D7 | RT D14 | After RT D10 | Before RT | RT D7 | RT D14 | After RT D10 |
| Autocorrelation*, † | 1.000 ± 0.000 | 0.874 ± 0.113 | 0.716 ± 0.190 | 0.579 ± 0.255 | 1.000 ± 0.000 | 0.995 ± 0.049 | 0.951 ± 0.084 | 0.908 ± 0.142 |
| Contrast* | 1.000 ± 0.000 | 2.026 ± 1.261 | 3.380 ± 2.425 | 4.468 ± 4.430 | 1.000 ± 0.000 | 1.116 ± 0.570 | 1.565 ± 0.784 | 1.792 ± 0.920 |
| Correlation | 1.000 ± 0.000 | 1.001 ± 0.005 | 1.003 ± 0.008 | 1.009 ± 0.008 | 1.000 ± 0.000 | 0.998 ± 0.007 | 1.000 ± 0.009 | 1.004 ± 0.008 |
| Cluster prominence | 1.000 ± 0.000 | 9.390 ± 27.491 | 10.022 ± 16.845 | 71.194 ± 187.39 | 1.000 ± 0.000 | 1.767 ± 2.244 | 2.028 ± 1.737 | 5.985 ± 6.946 |
| Cluster shade | 1.000 ± 0.000 | 3.540 ± 11.673 | 0.797 ± 16.016 | 13.359 ± 59.846 | 1.000 ± 0.000 | 1.392 ± 1.325 | 1.348 ± 1.070 | 2.478 ± 4.292 |
| Dissimilarity* | 1.000 ± 0.000 | 1.864 ± 0.956 | 2.800 ± 1.767 | 3.492 ± 3.035 | 1.000 ± 0.000 | 1.112 ± 0.545 | 1.499 ± 0.730 | 1.718 ± 0.839 |
| Energy | 1.000 ± 0.000 | 0.497 ± 0.279 | 0.330 ± 0.242 | 0.303 ± 0.319 | 1.000 ± 0.000 | 0.935 ± 0.362 | 0.747 ± 0.445 | 0.662 ± 0.461 |
| Entropy* | 1.000 ± 0.000 | 1.451 ± 0.457 | 1.746 ± 0.710 | 1.973 ± 1.099 | 1.000 ± 0.000 | 1.039 ± 0.321 | 1.233 ± 0.381 | 1.361 ± 0.417 |
| Homogeneity* | 1.000 ± 0.000 | 0.934 ± 0.055 | 0.889 ± 0.091 | 0.862 ± 0.134 | 1.000 ± 0.000 | 0.993 ± 0.033 | 0.966 ± 0.053 | 0.953 ± 0.055 |
| Maximum probability | 1.000 ± 0.000 | 0.584 ± 0.329 | 0.565 ± 0.864 | 0.670 ± 1.600 | 1.000 ± 0.000 | 0.931 ± 0.305 | 0.773 ± 0.405 | 0.721 ± 0.424 |
| Variance*, † | 1.000 ± 0.000 | 0.874 ± 0.112 | 0.716 ± 0.190 | 0.580 ± 0.255 | 1.000 ± 0.000 | 0.995 ± 0.049 | 0.951 ± 0.084 | 0.908 ± 0.142 |
| Sum average*, † | 1.000 ± 0.000 | 0.930 ± 0.065 | 0.831 ± 0.132 | 0.727 ± 0.199 | 1.000 ± 0.000 | 0.997 ± 0.026 | 0.974 ± 0.045 | 0.947 ± 0.088 |
| Sum variance*, † | 1.000 ± 0.000 | 0.856 ± 0.109 | 0.697 ± 0.191 | 0.540 ± 0.234 | 1.000 ± 0.000 | 0.994 ± 0.056 | 0.945 ± 0.092 | 0.899 ± 0.149 |
| Sum entropy* | 1.000 ± 0.000 | 1.412 ± 0.429 | 1.654 ± 0.640 | 1.836 ± 0.960 | 1.000 ± 0.000 | 1.035 ± 0.308 | 1.214 ± 0.362 | 1.335 ± 0.398 |
| Difference variance* | 1.000 ± 0.000 | 2.026 ± 1.261 | 3.380 ± 2.425 | 4.468 ± 4.430 | 1.000 ± 0.000 | 1.116 ± 0.570 | 1.565 ± 0.784 | 1.792 ± 0.920 |
| Difference entropy* | 1.000 ± 0.000 | 1.507 ± 0.511 | 1.915 ± 0.830 | 2.148 ± 1.258 | 1.000 ± 0.000 | 1.056 ± 0.333 | 1.281 ± 0.423 | 1.405 ± 0.443 |
| IMC 1* | 1.000 ± 0.000 | 0.921 ± 0.067 | 0.878 ± 0.100 | 0.882 ± 0.139 | 1.000 ± 0.000 | 0.986 ± 0.040 | 0.954 ± 0.061 | 0.951 ± 0.066 |
| IMC 2 | 1.000 ± 0.000 | 1.067 ± 0.090 | 1.089 ± 0.109 | 1.100 ± 0.131 | 1.000 ± 0.000 | 0.998 ± 0.061 | 1.031 ± 0.060 | 1.056 ± 0.074 |

*Abbreviations*: Before RT = before starting the radiotherapy; RT D$n$ = approximately $n$ days after the first day of the radiotherapy; After RT D$n$ = approximately $n$ days after the end of the radiotherapy; IMC = inverse measure of correlation.

*indicates statistical significances ($p < 0.05$) of both changes in the delta-radiomics within each group across time and differences in those for the interaction of the time and group with a two-way repeated measures analysis of variance (ANOVA).

†indicates statistical significances (p-value $< 0.05/3 = 0.0167$) of differences in the delta-radiomics between ipsilateral and contralateral breasts for all time points with a paired-t test or Wilcoxon signed rank test depending on the normality of data.

radiomic features, which were the *autocorrelation*, *homogeneity*, *variance*, *sum average*, *sum variance*, and *IMC 1*, showed statistically significant correlations ($p < 0.05$) to skin doses. The *sum variance* exhibited a significantly higher value of *r* compared with other radiomic features to skin doses. In the UV imaging mode, radiomic features with *p*-values less than 0.05 included the *variance*, and *sum variance*. Similar to the polarized imaging mode, the *sum variance* exhibited a significantly higher value of *r* compared with other radiomic features to skin doses. In the three imaging modes, the *r* values of radiomic features with *p*-values less than 0.05 were higher than 0.44.

## Correlations between delta-radiomics for all time points

The values of *r* and the corresponding *p*-values of the delta-radiomics of 'RT D7' to 'RT D14' and 'After RT D10' are shown in Table 6. Only the *r* values with *p*-values less than 0.05 are listed in Table 6. For this analysis, the delta-radiomics for ipsilateral breasts were used.

In the normal imaging mode, eight radiomic features for 'RT D7,' which included the *autocorrelation*, *cluster prominence*, *cluster shade*, *energy*, *maximum probability*, *variance*, *sum average*, and *sum variance*, exhibited statistically significant correlations with those at both 'RT D14' and 'After RT D10.' Among these, the *variance* indicated the strongest correlations at both 'RT D14' and 'After RT D10.' In the polarized imaging mode, all radiomic features with *p*-values less than 0.05 exhibited statistically significant correlations with both 'RT D14' and

**Table 5. Correlation coefficients ($r$) with corresponding $p$-values of Pearson's or Spearman's correlation test between skin dose values and delta-radiomics approximately 7 days after the first day of the radiotherapy ('RT D7').**

| | Normal | | Polarized | | UV | |
|---|---|---|---|---|---|---|
| | $r$ | $p$ | $r$ | $p$ | $r$ | $p$ |
| Autocorrelation | - | - | -0.540 | 0.005 | - | - |
| Contrast | - | - | - | - | - | - |
| Correlation | - | - | - | - | - | - |
| Cluster prominence | - | - | - | - | - | - |
| Cluster shade | - | - | - | - | - | - |
| Dissimilarity | - | - | - | - | - | - |
| Energy | -0.625 | 0.008 | - | - | - | - |
| Entropy | 0.515 | 0.032 | - | - | - | - |
| Homogeneity | -0.527 | 0.034 | -0.479 | 0.029 | - | - |
| Maximum probability | -0.545 | 0.009 | - | - | - | - |
| Variance | - | - | -0.540 | 0.006 | -0.485 | 0.038 |
| Sum average | - | - | -0.542 | 0.024 | - | - |
| Sum variance | - | - | -0.569 | 0.005 | -0.590 | 0.048 |
| Sum entropy | - | - | - | - | - | - |
| Difference variance | - | - | - | - | - | - |
| Difference entropy | - | - | - | - | - | - |
| IMC 1 | - | - | -0.491 | 0.032 | - | - |
| IMC 2 | - | - | - | - | - | - |

*Abbreviations*: UV = ultraviolet; IMC = inverse measure of correlation.

'After RT D10,' except for the *autocorrelation*, *contrast*, *dissimilarity*, *homogeneity*, *variance*, *sum variance*, *difference variance*, and *IMC 1*. The *cluster prominence* and *sum entropy* had significantly higher $r$ values than other values. In the UV imaging mode, every value of $r$, except for the *IMC 1*, was statistically significant and exhibited $p$-values less 0.05. Among these, the *cluster prominence* showed statistically significant correlations to those at both 'RT D14' and 'After RT D10.' Statistically significant correlations were more frequently observed in the UV imaging mode than other modes.

## Discussion

In this study, we applied a radiomics approach to skin surface images acquired by a skin analysis device to objectively assess the acute radiation dermatitis in patients undergoing radiotherapy for breast cancer. The 18 radiomic features were calculated from images in the normal, polarized, and UV modes before, during, and after radiotherapy. To test the performance of radiomic features as indicators of radiodermatitis evaluation, we assessed the statistically significant changes and differences in delta-radiomics within and between groups (ipsilateral and contralateral breasts) over time and correlations between the delta-radiomics and the RTOG score or skin dose. Furthermore, correlations between the delta-radiomics at 'RT D7' to those at 'RT D14' and 'After RT D10' for the ipsilateral breast were analysed to evaluate whether the degree of acute radiation dermatitis can be predicted in advance using the feature.

Fig 2 shows that the skin surface images and the patterns of the corresponding GLCMs in the three imaging modes were changed as the skin was damaged by radiation. Accordingly, most of the delta-radiomics for the ipsilateral breasts showed increasing or decreasing tendencies, exhibiting better performance or correlations in all statistical analyses. Of these, the

**Table 6. Correlation coefficients (*r*) with corresponding *p*-values of Pearson's or Spearman's correlation test of delta-radiomics approximately 7 days after the first day of the radiotherapy ('RT D7') to those on 'RT D14' and approximately 10 days after the end of the radiotherapy ('After RT D10').**

| | Normal | | | | Polarized | | | | UV | | | |
|---|---|---|---|---|---|---|---|---|---|---|---|---|
| | RT D14 | | After RT D10 | | RT D14 | | After RT D10 | | RT D14 | | After RT D10 | |
| | r | p | r | p | r | p | r | p | r | p | r | p |
| Autocorrelation | 0.709 | <0.001 | 0.631 | 0.003 | - | - | 0.462 | 0.040 | 0.765 | <0.001 | 0.723 | <0.001 |
| Contrast | - | - | 0.629 | 0.003 | - | - | - | - | 0.728 | <0.001 | 0.472 | 0.036 |
| Correlation | - | - | - | - | 0.537 | 0.015 | 0.830 | <0.001 | 0.652 | 0.002 | 0.567 | 0.009 |
| Cluster prominence | 0.470 | 0.036 | 0.485 | 0.030 | 0.942 | <0.001 | 0.794 | <0.001 | 0.944 | <0.001 | 0.943 | <0.001 |
| Cluster shade | 0.479 | 0.033 | 0.518 | 0.019 | 0.574 | 0.008 | 0.457 | 0.043 | 0.618 | 0.004 | 0.933 | <0.001 |
| Dissimilarity | - | - | 0.535 | 0.015 | 0.616 | 0.004 | - | - | 0.726 | <0.001 | 0.459 | 0.042 |
| Energy | 0.615 | 0.020 | 0.693 | 0.003 | 0.775 | <0.001 | 0.609 | 0.004 | 0.584 | 0.007 | 0.449 | 0.047 |
| Entropy | - | - | - | - | 0.839 | <0.001 | 0.770 | <0.001 | 0.871 | <0.001 | 0.667 | 0.001 |
| Homogeneity | - | - | - | - | 0.561 | 0.010 | - | - | 0.743 | <0.001 | 0.509 | 0.022 |
| Maximum probability | 0.625 | 0.003 | 0.513 | 0.021 | 0.592 | 0.006 | 0.561 | 0.010 | 0.636 | 0.003 | 0.509 | 0.022 |
| Variance | 0.710 | <0.001 | 0.631 | 0.003 | - | - | 0.462 | 0.040 | 0.764 | <0.001 | 0.723 | <0.001 |
| Sum average | 0.706 | 0.001 | 0.621 | 0.003 | 0.789 | 0.010 | 0.762 | 0.040 | 0.743 | <0.001 | 0.698 | 0.001 |
| Sum variance | 0.700 | 0.001 | 0.632 | 0.003 | - | - | 0.463 | 0.040 | 0.761 | <0.001 | 0.716 | <0.001 |
| Sum entropy | 0.591 | 0.006 | - | - | 0.869 | <0.001 | 0.828 | <0.001 | 0.889 | <0.001 | 0.682 | 0.001 |
| Difference variance | - | - | 0.629 | 0.003 | - | - | - | - | 0.728 | <0.001 | 0.472 | 0.036 |
| Difference entropy | - | - | 0.486 | 0.030 | 0.618 | 0.004 | 0.502 | 0.024 | 0.826 | <0.001 | 0.594 | 0.006 |
| IMC 1 | - | - | - | - | - | - | 0.582 | 0.007 | 0.712 | <0.001 | - | - |
| IMC 2 | - | - | - | - | 0.823 | <0.001 | 0.803 | <0.001 | 0.864 | <0.001 | 0.780 | <0.001 |

*Abbreviations*: UV = ultraviolet; RT D*n* = approximately *n* days after the first day of the radiotherapy; After RT D*n* = approximately *n* days after the end of the radiotherapy; IMC = inverse measure of correlation

performance of the *energy* in the normal mode and the *sum variance* in the polarized and UV modes were better than others in general.

The normal mode in the skin analysis device could analyse skin pores. As shown in Fig 2 (A), it was observed that the skin surface became redder and darker over time on the side of the breast cancer treatment. The results of this study correspond well with those obtained in earlier studies [19, 27, 28, 41]. Previous studies reported that the skin reaction for irradiated breasts exhibited higher a* values (reddish) and lower L* values (darker); these values were measured by a spectrophotometer during radiotherapy. Momm *et al.* showed strong correlations to the radiation dose using spectrophotometry. By analysing the spatial distribution of the GLCM intensities in Fig 2(A), it can be observed that as the skin was damaged by radiation, the peak of the high intensities of GLCMs tended to gradually disappear. These intensities were subsequently shifted toward the lowest elements of the GLCMs. It was demonstrated that intensity homogeneity of the skin surface images deteriorated, and the colour tone of the skin became darker as the radiodermatitis progressed. The *energy* for the normal imaging mode, which exhibited the best performance for assessing acute radiation dermatitis, had decreasing tendencies over time because it is a measure of homogeneity of an image. A higher value for this feature indicates that the intensity in an image varies less [42].

The polarized mode in the skin analysis device could evaluate the melanin pigmentation of the basal layer of the epidermis. Fig 2(B) shows that the skin pigmentation in ipsilateral breasts increased and peaked 'After RT D10.' The results of this study are similar to those of Hu *et al.* [29]. Their study reported that the skin pigmentation of irradiated breasts increased, owing to radiation damage, compared with that of unirradiated breasts. The intensities in the GLCM

distributions generated from skin surface images in the polarized mode were gradually spread out, and these intensities then shifted toward the lowest elements of the GLCMs, demonstrating that there was inhomogeneity and darkness in the images. The *sum variance* typically exhibits high values when the intensities of the GLCM distributions are gathered in the lowest and highest elements [42]. For this reason, the values of the *sum variance* decreased over time in this study.

The UV mode in the skin analysis device was typically utilised to evaluate skin sebum; however, it also showed melanin accumulations below the skin surface. In this study, the skin surface images in the UV mode continued to darken from the periphery inward, toward the centre of the images, over time. This demonstrated that the melanin accumulated, owing to radiation damage, as shown in Fig 2(C). It was difficult to evaluate radiodermatitis using skin sebum. Hu *et al.* also showed that there was no significant change in skin sebum in both ipsilateral and contralateral breasts before and after radiotherapy [29]. Unlike other imaging modes, the periphery of the images in the UV mode appeared optically darker than the centre, owing to non-uniformity in UV intensity. For this reason, we cropped the circular ROI of the original images obtained in UV mode. Other images in the normal and polarized mode were also cropped in the same manner to calculate the radiomic features. The intensities in the GLCM distributions generated from skin surface images in the UV mode exhibited a pattern similar to those in the polarized mode. Therefore, the *sum variance* also had decreasing tendencies, providing the best performance for both polarized and UV imaging modes.

Before initiating the experiment, the skin analysis device utilized in this study was evaluated for variability and reproducibility of radiomic features under various conditions and environments. When measured more than five times in each of the three imaging modes under identical environmental conditions, the deviations in the radiomic feature values remained within 0.5%, demonstrating the device's good reproducibility. Even in extreme environmental scenarios, such as a dark room with approximately 100 lux, the device exhibited small deviations ($< 0.5\%$) from radiomic feature values. The minimal effect of conditions and environments on this device can be attributed to its independent light source and the method of closely adhering to the patient's skin to acquire an image. By conducting various simulations, we established an optimal protocol for skin measurement to minimize errors and uncertainty in the experiment. Consequently, the bias in the correlation between skin damage and feature was reduced to the greatest extent possible. Fig 3 presents the changes in the delta-radiomics of the representative features for ipsilateral and contralateral breasts over time in the three imaging modes. As

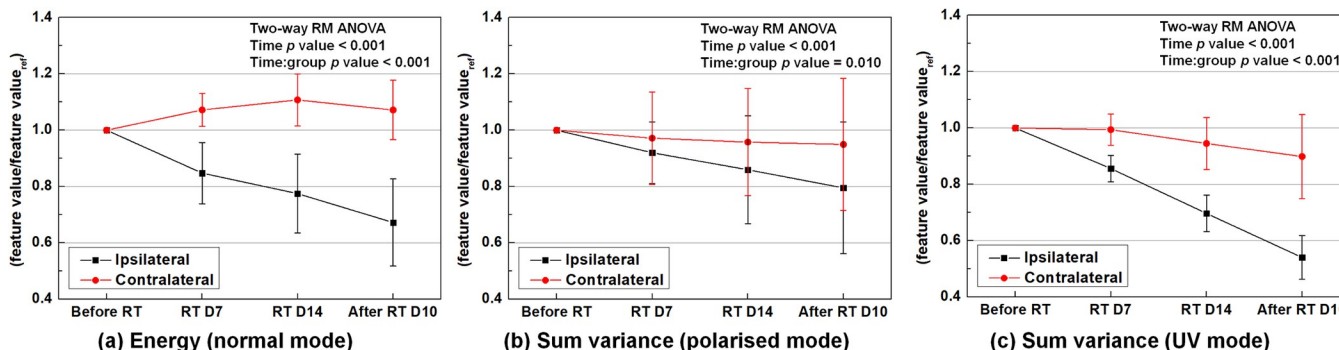

**Fig 3. Comparison of the delta-radiomics of the representative features for the ipsilateral and contralateral breasts over time in the three imaging modes.** The representative features are the *energy* for the (a) normal imaging mode and the *sum variance* for the (b) polarized and (c) UV imaging modes. The time points for the measurements were before starting the radiotherapy ('Before RT'), approximately 7 days after the first day of the radiotherapy ('RT D7'), on 'RT D14,' and approximately 10 days after the end of the radiotherapy ('After RT D10').

mentioned above, these radiomic features showed the statistical significance of the two-way repeated measures ANOVA test. In the unirradiated breasts, there were unpredictable tendencies in the *energy* in the normal mode. There were decreasing tendencies in the *sum variance* in both the polarized and UV modes; however, these showed a much smaller decrease compared to the irradiated breasts. This may be explained by the fact that radiation that was used to treat breast cancer was scattered or penetrated to the contralateral breast, thereby causing damage. Hu *et al.* reported similar results, showing an increase in pigmentation for the contralateral breast after radiotherapy [29]. However, with our small cohort size, the results in the contralateral breasts could be indicative of overfitting, showing the large variance of the delta-radiomics. It was demonstrated that the scattered or penetrated radiations to the contralateral breasts should not cause enough damage to skin to induce acute radiodermatitis. The number of unknown covariates might exceed the sample size (metabolism, menstrual cycle, effect of clothing that patients should wear before imaging), rendering any statistical analysis useless.

Several studies have addressed the limitation of these subjective methods, which create several uncertainties. They demonstrated that these methods could not represent patient-reported breast symptoms, including pain, itching, tightness, and local heat [43]. The subjective assessments were affected by intra- and inter-evaluator variations, and were considered less suitable for close scrutiny [15, 22, 31]. Although these various studies highlighted the limitations of the clinician-assessed scoring criteria, the performances of various skin biophysical parameters as indicators for assessing radiodermatitis were evaluated by analysing the correlations between the scoring criteria and the parameters. In this study, we used the RTOG scoring criteria, and all enrolled patients were evaluated as grade 1 on 'RT D14' and 'After RT D10.' For this reason, our study could not show the correlations between the RTOG scoring and the radiomic feature ratios for different time points. The small number of samples in our study noted did not influence the fact that the severity of radiodermatitis is limited to grade 1 or 2. Several institutions have reported that the severity of acute radiation dermatitis could be reduced when IMRT or FIF techniques were used due to minimized unwanted radiation dose inhomogeneity in the breasts [44–46]. In addition, Cordoba *et al.* showed that the smaller body mass index (BMI) and breast size, the lower the radiation-induced skin damage was evaluated as grade 1 or 2 [47]. Overall, we enrolled Korean female patients with normal BMI and small breasts and used IMRT or FIF techniques without boost irradiation, which resulted in assessing all patients within grade 1, regardless of the cohort size [47, 48].

We presented strong correlations between the skin doses and radiomic feature ratios on 'RT D7.' The skin doses measured by OSLDs were reported as an absorbed dose at a skin depth of 0.4 mm, and this data was then utilized to assess the effect of radiation on the basal layer of the skin epidermis [49]. Thus, the severity of acute radiation dermatitis can vary within the same grade, showing the limitations of clinician-assessed scoring criteria. Because there is a significant relationship between the severity of radiodermatitis and the absolute doses delivered to the patients' skin [6, 7], we could evaluate the severity of acute radiation dermatitis objectively using representative radiomic features (on 'RT D7'), regardless of the clinical-assessed scoring criteria. Furthermore, the representative radiomic features exhibited strong correlations to those on 'RT D14' and 'After RT D10,' which means that radiation-induced skin damage during and after radiotherapy can be predicted in advance. Using a radiomics approach to skin surface images, it is possible to subdivide the severity of acute radiation dermatitis even within the same grade, which can play a very important role in terms of developing and evaluating the topical agents for acute radiodermatitis.

In future work, this radiomic approach for the evaluation of radiodermatitis will be further applied to patients undergoing radiotherapy for head and neck cancer, where acute radiation dermatitis is predominantly observed. Because patients over grade 2+ were not included, we

were unable to evaluate the comprehensive severity of radiodermatitis, which is a limitation of this study. Further, the small sample size belonged to one pure race (Asian patients) and lack of external validation are obvious limitations of this study. Spurious correlations could be expected because of insufficient small cohort size and large feature numbers. Our study, being a pilot study, tried to evaluate whether the radiomics approach has the potential to objectively evaluate the severity of radiodermatitis quickly and efficiently. However, independent external validation is essential to determine whether the delta-radiomics obtained from topical images of skin surface can be considered as a generalizable model for evaluating the radiation-induced skin damage by ruling out potential overfitting in statistical modelling. Additionally, it was reported that the expression of radiodermatitis was influenced by race and ethnicity, demonstrating that severe skin damage from radiation was more commonly observed in individuals with black skin than in those with white skin [50]. Therefore, further research using a large number of samples and various severities of radiodermatitis, various treatment sites, and various races from multiple institutions will be performed in the future. Skin doses will be measured by using various dosimeters and calculated by radiation treatment planning systems.

As many studies have shown that radiomic features can vary when calculated under different imaging processing and software programs, the Image Biomarker Standardisation Initiative (IBSI) guidelines offer a consensus for standardized methods for calculating all radiomic features [51]. Radiomics programs should be compliant with IBSI guidelines for improved reproducibility [51]. Although open-source code in MATLAB software used in our study to calculate radiomic features was compliant with the IBSI guidelines, the total number of the calculated features was very limited [18]. In future, more than 300 features would be calculated using several publicly available software program (e.g. Pryadiomics, SERA, etc.). In addition to the hand-crafted radiomic features utilised in our study, quantitative radiomic features extracted using a convolutional neural network can be used in supervised machine learning to predict the radiodermatitis [25].

## Conclusions

A radiomics approach to skin surface images acquired by a skin analysis device was applied to objectively assess acute radiation dermatitis in patients undergoing radiotherapy for breast cancer. In general, the *energy* for the normal mode and *sum variance* for polarized and UV modes exhibited better performance than others in evaluating the radiation-induced skin damage. Using the ratios of the *energy* and *sum variance* on 'RT D7,' radiodermatitis severity which cannot be detected by a human observer during and after radiotherapy can be assessed, which assists in its appropriate management.

## Supporting information

**S1 Checklist.**
(PDF)

**S1 Table. Skin dose values measured using a nanoDot optically stimulated luminescent dosimeter (OSLD) for ipsilateral breasts.** Four measurement points were placed on the upper, lower, inner, and outer sides of the ipsilateral breasts.
(DOCX)

**S2 Table. *P*-values of a two-way repeated measures analysis of variance (ANOVA) in the three imaging modes.**
(DOCX)

**S3 Table. *P*-values of a paired-t tests or Wilcoxon signed rank test in the three imaging modes.**
(DOCX)

## Author Contributions

**Conceptualization:** So-Yeon Park, Jung-in Kim, Jin Ho Kim.

**Data curation:** So-Yeon Park, Jong Min Park, Chang Heon Choi.

**Formal analysis:** So-Yeon Park, Jung-in Kim, Ji Hyun Chang.

**Funding acquisition:** So-Yeon Park, Jung-in Kim.

**Investigation:** So-Yeon Park, Jung-in Kim, Chang Heon Choi, Minsoo Chun.

**Methodology:** Jong Min Park, Jung-in Kim.

**Resources:** Minsoo Chun, Ji Hyun Chang, Jin Ho Kim.

**Software:** So-Yeon Park.

**Supervision:** Jung-in Kim, Jin Ho Kim.

**Validation:** So-Yeon Park, Jong Min Park, Ji Hyun Chang.

**Visualization:** So-Yeon Park.

**Writing – original draft:** So-Yeon Park.

**Writing – review & editing:** Jong Min Park, Jin Ho Kim.

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
