## [Decision Letter · Decision Letter 0]

23 Jun 2023

PONE-D-23-06647Quantitative radiomics approach to assess acute radiation dermatitis in breast cancer patientsPLOS ONE

Dear Dr. Kim,

Thank you for submitting your manuscript to PLOS ONE. After careful consideration, we feel that it has merit but does not fully meet PLOS ONE’s publication criteria as it currently stands. Therefore, we invite you to submit a revised version of the manuscript that addresses the points raised during the review process.

We look forward to receiving your revised manuscript.

Kind regards,

Daniele Ugo Tari, M.D.

Academic Editor

PLOS ONE

“This study was supported by Basic Science Research Program through the National Research Foundation of Korea (NRF) funded by the Ministry of Education (NRF- 2017R1D1A1B03036093) and by a VHS Medical Center Research Grant, Korea (grant number: VHSMC 21035).”

“This study was supported by Basic Science Research Program through the National Research Foundation of Korea (NRF) funded by the Ministry of Education (NRF- 2017R1D1A1B03036093) and by a VHS Medical Center Research Grant, Korea (grant number: VHSMC 21035).”

“This study was supported by Basic Science Research Program through the National Research Foundation of Korea (NRF) funded by the Ministry of Education (NRF- 2017R1D1A1B03036093) and by a VHS Medical Center Research Grant, Korea (grant number: VHSMC 21035).”

Reviewers' comments:

Reviewer's Responses to Questions

**Comments to the Author**

1. Is the manuscript technically sound, and do the data support the conclusions?

Reviewer #1: Partly

Reviewer #2: Yes

Reviewer #3: Yes

2. Has the statistical analysis been performed appropriately and rigorously? 

Reviewer #1: No

Reviewer #2: Yes

Reviewer #3: Yes

3. Have the authors made all data underlying the findings in their manuscript fully available?

Reviewer #1: Yes

Reviewer #2: Yes

Reviewer #3: Yes

4. Is the manuscript presented in an intelligible fashion and written in standard English?

Reviewer #1: Yes

Reviewer #2: Yes

Reviewer #3: Yes

5. Review Comments to the Author

Reviewer #1: Dear authors,

Thank you for presenting to me the article “Quantitative radiomics approach to assess acute radiation dermatitis in breast cancer”. The idea of usings quantitative radiomics in the assessment of radiodermatitis is interesting.

However, the article lacks a clear research question and structure for answering this question. This is particular important since 1728 radiomic features per patient (line 181) are used. Moreover, one should be aware of the multiple comparison problem, e.g. finding significance by chance after doing lots of testing.

Other comments/ideas:

- Rule 289: Why were the ipsilateral breasts used?

- Practicality of the device? Experience? Errors?

- What is the variability of radiomics when repeating the experiment at the same time, same patient and same location and same imaging mode?

- What is the clinical benefit of using this device?

- Can this device assist radiotherapists/oncologists when assessing radiodermatitis? (without the use of radiomics)

Reviewer #2: Hi

Overall this pilot study is conducted impressively from the point of view of clarity, objectivity, quantification, writing, presentation and new information.

However, I have few comments which needs response

1. Is skin dose measurement in this study using nano dot and OSLD is gold standard? And calibration

details?

2. Was the BMI taken into account (size the breast) and correlation with skin dose would be interesting?

3. Patient characteristics and Tumor characteristics does it have any effect on skin dose (eg. Age, color complexion (fair, brown and dark)?

4. Was the imaging dose( both CT and patient positioning verification dose)?

Thanks

Reviewer #3: This is a well-carried out and well-written research on acute radiation dermatitis in patients undergoing breast cancer radiotherapy. It showed meticulous attention to details.

A pertinent question that may be addressed include:

1. Is there a reason or reasons why scar boost is not given in these patients? Scar boost is standard in many breast conserving radiotherapy in many radiotherapy treatment guidelines. Could this impact future risk of disease recurrence on the irradiated breasts in these patients?

2. Detailed histopathological characterization of the patient's breast surgical specimens not provided, viz-a-viz resection margin status, DCIC with microinvasion [DCIS-M] etc. This could explain the lack of radiotherapy boost to tumor bed.

Please clarify these issues when you respond to the reviewers' comments

6. PLOS authors have the option to publish the peer review history of their article (what does this mean?). If published, this will include your full peer review and any attached files.

Reviewer #1: No

Reviewer #2: No

Reviewer #3: No

---

## [Author Response · Author response to Decision Letter 0]

20 Aug 2023

We appreciate the reviewers’ time and effort in reviewing our manuscript. The quality of the manuscript has been substantially improved with the aid of the reviewer's valuable comments. The revised version of our manuscript has addressed the reviewer's concerns as attached "Response_to_Reviewers".

---

## [Editor Report · Decision Letter 1]

5 Oct 2023

Quantitative radiomics approach to assess acute radiation dermatitis in breast cancer patients

PONE-D-23-06647R1

Dear Dr. Kim,

We’re pleased to inform you that your manuscript has been judged scientifically suitable for publication and will be formally accepted for publication once it meets all outstanding technical requirements.

Kind regards,

Daniele Ugo Tari, M.D.

Academic Editor

PLOS ONE
---

## [Editor Report · Acceptance letter]

17 Oct 2023

PONE-D-23-06647R1 

Quantitative radiomics approach to assess acute radiation dermatitis in breast cancer patients 

Dear Dr. Kim:

I'm pleased to inform you that your manuscript has been deemed suitable for publication in PLOS ONE. Congratulations! Your manuscript is now with our production department. 

Kind regards, 

on behalf of

Dr. Daniele Ugo Tari 

Academic Editor

PLOS ONE